# Meropenem/Vaborbactam—A Mechanistic Review for Insight into Future Development of Combinational Therapies

**DOI:** 10.3390/antibiotics13060472

**Published:** 2024-05-21

**Authors:** Trae Hillyer, Woo Shik Shin

**Affiliations:** 1Department of Pharmaceutical Sciences, Northeast Ohio Medical University, Rootstown, OH 44272, USA; thillyer@neomed.edu; 2University Hospital and Northeast Ohio Medical University Scholarship Program, Rootstown, OH 44272, USA

**Keywords:** beta-lactam antibiotics, combination therapy, meropenem, vaborbactam

## Abstract

Beta-lactam antibiotics have been a major climacteric in medicine for being the first bactericidal compound available for clinical use. They have continually been prescribed since their development in the 1940s, and their application has saved an immeasurable number of lives. With such immense use, the rise in antibiotic resistance has truncated the clinical efficacy of these compounds. Nevertheless, the synergism of combinational antibiotic therapy has allowed these drugs to burgeon once again. Here, the development of meropenem with vaborbactam—a recently FDA-approved beta-lactam combinational therapy—is reviewed in terms of structure rationale, activity gamut, pharmacodynamic/pharmacokinetic properties, and toxicity to provide insight into the future development of analogous therapies.

## 1. Introduction

The first beta-lactam antibiotic, penicillin, became available in the 1940s, and it proved to be a life-saving treatment for what would be considered a minor infection today [1]. For its cardinal role in medicine, the Nobel Prize was awarded to its developers; Fleming, Chain, and Florey, in 1945 [2]. The discovery, isolation, and development of penicillin initiated the research leading to the discovery of many other fermentation products used as antibiotics [2]. To no surprise, beta-lactams have become the largest class of antibiotics today, covering 65% of the market, and are the inspiration for modern treatments like meropenem/vaborbactam [3]. The mechanism of action of beta-lactam antibiotics involves covalent inhibition of a peptidoglycan transpeptidase in the bacterial cell wall, known as penicillin-binding protein [4,5]. Penicillin-binding protein catalyzes the crosslinking of peptidoglycan layers within the cell wall, providing structure and integrity to the cell [6]. When beta-lactams bind to penicillin-binding protein, the structural integrity of peptidoglycan cross-linkage is compromised, and the cells are susceptible to lysis by osmotic pressure [7]. The specificity of beta-lactams to their non-eukaryotic target and well-established pharmacokinetics/pharmacodynamics renders them the primary choice for physicians when treating bacterial infections [8].

Beta-lactams consist of five separate categories based on their chemical structure, but consistent within them all is the beta-lactam ring composed of three carbons and one nitrogen [1]. These include penicillins, cephalosporins, carbapenems, and monobactams [5]. Figure 1 shows the structural differences between the different classes of beta-lactams.

With beta-lactams being the most commonly prescribed antibiotic, resistance to treatment was bound to occur. In fact, Alexander Fleming predicted this himself due to the “era of overuse” [9]. As warned, widespread resistance to penicillin became common within ten years of its introduction to the market [10]. Resistance to beta-lactams, specifically, is induced primarily through the expression of hydrolytic enzymes known as beta-lactamases [11,12]. Beta-lactamases utilize a serine residue as a nucleophile to attack the carbonyl carbon of the beta-lactam ring, leading to the hydrolysis and deactivation of the beta-lactam antibiotic [13]. To overcome beta-lactamase-induced resistance to beta-lactam antibiotics, the combinational administration of a beta-lactam with a beta-lactamase inhibitor to prevent enzymatic hydrolyzation was first demonstrated with amoxicillin and clavulanic acid in 1981 [14]. The development of a new class of potent β-lactamase inhibitors to address the existing β-lactam antibiotic resistance offers the greatest opportunity for maximizing the efficacy of combination antimicrobial therapy, aiming to preserve the potency of existing β-lactam antibiotics. There are currently six beta-lactamase inhibitors available for use in combinational therapy, as shown in Figure 2 [15].

From Figure 2, notice that clavulanate, sulbactam, and tazobactam all contain beta-lactam rings, whereas avibactam, relebactam, and vaborbactam do not. The beta-lactam-containing inhibitors are effective against primarily class A and C serine beta-lactamases by forming sterically unfavorable acyl-enzyme interactions [15,16]. The latter beta-lactamase inhibitors, the non-beta-lactams, covalently bind to a serine residue consistent in the active site of beta-lactamases, permanently inactivating them via suicide inhibition [15]. The diazabicyclooctane inhibitors, avibactam and relebactam, were meant to be broad-spectrum inhibitors of class A, C, and some D beta-lactamases and have proven to be efficacious, even inadvertently, against the very prevalent Klebsiella pneumoniae carbapenemase (KPC) [17]. However, there have already been reports of reduced potency to these diazabicyclooctane inhibitors due to mutations in the target enzymes both clinically and in vitro [16,18,19,20]. Reports of treatment failure and relapse due to resistance developing within the duration of therapy have also been reported [17,21,22]. Vaborbactam, the only beta-lactamase inhibitor that is a cyclic boronic acid, was also developed to be a broad-spectrum inhibitor with specific potency for KPC in mind [17,23].

## 2. Structural Development of Vaborbactam

Boronic acids, like vaborbactam, have been previously shown to have inhibitory activity against serine proteases but were first investigated for their activity against beta-lactamases by Oxford University in the late 1970s [23,24,25,26]. The cyclic boronic acid structure of vaborbactam was inspired by the work of Ness et al., who hypothesized that the phenolic hydroxyl group of their lead compound might induce the formation of a cyclic boronate ester upon binding [27]. Hecker et al. thought that a cyclic boronate ester would ideally constrain the compounds into the preferred conformation for docking while also providing higher selectivity for beta-lactamases, rather than other mammalian serine proteases, which form a more linear transition state profile [26]. In silico modeling of non-covalent interactions was conducted to determine which compounds had a high affinity for the active site, which would ultimately allow for the rapid formation of the covalent interaction [26]. The highest-ranking inhibitor from in silico docking is shown in Figure 3. The lead compound was tested with multiple substituents, including N-acetyl, phenylacetyl, 2-thienyl acetyl, aminothiazole, aminopyridyl, and various hydroxyl group configurations, before finding the most potent confirmation, a thiophene, shown in Figure 3 [26].

The final product after the addition of the thiophene to the cyclic boronate ester, vaborbactam, confirmed its binding mechanism through X-ray diffraction. The exact binding mechanism against KPC-2 is shown in Figure 4 (PDB ID: 6V7I) [28].

As previously stated, vaborbactam was developed as a broad-spectrum inhibitor, but KPC beta-lactamases were of specific interest. Vaborbactam achieves nanomolar *K_i_* values for KPC by interactions formed within the oxyanion hole due to its non-linear, cyclic structure [28]. Vaborbactam is unique in this manner because it is the only inhibitor that is able to form an additional hydrogen bond with Thr237 via the exocyclic hydroxyl group [28,29,30]. This additional bond is consistent in complexes with CTX-M-14, but threonine is replaced by a serine [28]. This unique interaction of vaborbactam could be utilized in future inhibitor development as it seems to contribute to the high affinity and broad-spectrum activity of vaborbactam. Table 1 shows the inhibition constants of vaborbactam against various beta-lactamases.

Vaborbactam was tested with a range of beta-lactams including cephalosporins and monobactams but proved most effective in combination with meropenem, a carbapenem [23]. The physical and chemical properties of meropenem and vaborbactam can be seen in Table 2.

Meropenem was developed in the 1980s as a “me better” drug to the former a N-formimidoylthienamycin derivative, imipenem [35]. Imipenem was rapidly hydrolyzed by dehydropeptidase-I (DHP-1) in the renal tubules [35]. This led to reduced absorption into the urinary tract, reducing efficacy in urinary tract infections while also causing proximal tubular necrosis when given in high doses to achieve sufficient concentrations to treat cognate infections [35]. Meropenem was chemically optimized to avoid the issues of imipenem, which can be seen in Figure 5 by the structure–activity relationships as described by Drusano [35].

The C_1_ methyl group improved the stability of meropenem in DHP-1, allowing for higher concentrations of the drug to reach the urinary tract [35]. A recently approved combinational therapy, Recarbio, includes imipenem (carbapenem), relebactam (diazabicyclooctane beta-lactamase inhibitor), and cilistatin (DHP-1 inhibitor) [36]. The critical advantage of the meropenem/vaborbactam combinational therapy, known as Vaborem, is that there is no need to include a DHP-1 inhibitor because of the stability the C_1_ methyl group provides in DHP-1 [35,37,38]. Clinical trials show that 60–80% of meropenem reaches the urinary tract without combining it with a DHP-1 inhibitor due to resistance to DHP-1 provided by the C_1_ methyl group [37]. Cilistatin, a DHP-1 inhibitor, is given at a 1:1 ratio to imipenem in Recarbio; however, pharmacokinetic data shows that they are eliminated at different rates, meaning that the dosages would have to be adjusted for the continued treatment to be safe [39]. Additionally, the C_2_ pyrrolidine-3-thiol substituent increases the activity in gram-negative pathogens while also allowing for greater tolerance in the central nervous system [35].

### Clinical Use and PD/PK Data

The combination of meropenem and vaborbactam was approved by the FDA in 2017 for the treatment of patients aged 18 years and older with complicated urinary tract infections, including pyelonephritis, caused by *Enterobacteriaceae* [40]. Although this is the only FDA-approved use of the combinational therapy, it will likely be used to treat multi-drug resistant infections acquired nosocomialy [41]. The meropenem/vaborbactam combination should be administered every 8 h by intravenous infusion over 3 h for up to 14 days [40]. Meropenem has time-dependent bactericidal activity; therefore, the best indicator for dosage is the level of free drug that exceeds the MIC, also known as %T_>ƒMIC_ [42]. With the FDA-approved dosage, free meropenem levels should exceed 8 mg/L for at least 40% of the interval [42]. Data from the studies below show that this concentration was maintained. Pharmacokinetic data from 91 non-infected patients in phase 1 studies and from 322 infected patients in phase 3 trials can be seen in Table 3 and Table 4, respectively. Importantly, the pharmacokinetics of meropenem and vaborbactam are well correlated, allowing for simplistic dosage regimes.

Data from Trang et al., shown in Table 3, indicates that both meropenem and vaborbactam are primarily cleared by the kidneys (CLR) [43]. Another study done by Rubino et al. showed that the simultaneous administration of these two drugs does not affect plasma pharmacokinetics or renal clearance [44]. This same study showed that 47–64% of meropenem is excreted in the urine unchanged, and 81–95% of vaborbactam was excreted in the urine unchanged over the same time period, further validating the primary clearance route, indicating there are minimal, if any, toxic metabolites [44].

Rubino et al. also evaluated the safety and tolerability of vaborbactam and meropenem in combination and individually. Plasma pharmacokinetic exposure measurements (C_max_, AUC_0–t_, and AUC_0–inf_) all validated that exposure to vaborbactam or meropenem alone, or in combination, was no different, providing credibility of no drug-drug interactions [44]. Adverse events of vaborbactam and meropenem in combination mimic that of meropenem given independently [44,45,46,47]. A list of the most common adverse events can be seen in Table 5.

Notable differences from Table 5 include a higher percentage of headaches and injection site reactions that occurred when vaborbactam was included. Although higher, the relative trend of adverse events is relatively unchanged and mild. Hepatoxicity of meropenem has been reported in 1–6% of recipients via elevated serum aminotransferase levels when given for up to 14 days [48]. However, the elevated serum levels are usually transient, mild, and asymptomatic, with dosage adjustments rarely required [48]. There have also been rare cases of cholestatic jaundice linked to meropenem treatment [48]. There has been one case reported indicating that meropenem treatment induced vanishing vile duct syndrome and one other case where a patient experienced acute, generalized exanthematous pustulosis [49,50]. Aside from these rare incidences, meropenem is generally a well-tolerated treatment. There have not been much data published demonstrating the toxicity of vaborbactam alone, but it also has been well tolerated clinically, most likely due to its design. As previously mentioned, its cyclic structure reduces its affinity for mammalian serine proteases, which have a more linear active site [26]. The IC_50_ values of vaborbactam for mammalian serine proteases are shown in Table 6. The specificity for the target enzyme, and not adverse enzymes, is key to its tolerability and minimal side effects.

## 3. Discussion and Conclusions

Meropenem/vaborbactam is a unique combinational therapy because the structure of its inhibitor is boronic acid. For this reason, it can be expected to raise new ideas in the structural development of future combinational therapies. As shown, vaborbactam has broad-spectrum activity against all serine beta-lactamases but lacks affinity for the class B, metallo-beta-lactamases. There are currently no FDA-approved combinational therapies for class B-expressing pathogens [12]. Many are in development; however, they all are relatively specific to a few of the enzymes in this class and lack affinity for serine beta-lactamases [51]. The development of a beta-lactamase inhibitor that is effective against the entire spectrum of beta-lactamases would be an entirely new treatment option. Additionally, most pathogens carrying metallo-beta-lactamases also carry serine beta-lactamses [52,53,54,55]. Therefore, without combating both classes, a metallo-beta-lactamase inhibitor will most likely be ineffective, as the serine beta-lactamases would still induce resistance. The clinical use of meropenem/vaborbactam is still relatively new; therefore, its clinical efficacy against metallo-beta-lactamases in combination with other inhibitors is still unknown [56]. Another strategy could be adding a class B specific inhibitor to an already approved combination therapy for serine beta-lactamases, such as meropenem/vaborbactam. This development process could potentially be fast-tracked to approvable if added to an already approved therapy as their therapeutic potential is already well documented. This strategy was shown to be successful using meropenem/vaborbactam/aztreonam combination to treat New Delhi beta-lactamase-expressing *Klebsiella pneumoniae* [57,58]. The established pharmacokinetics and pharmacodynamics of beta-lactams make them ideal for use in combinational therapies. They have been proven to be efficacious when they can reach their target, and allowing them to reach said target is indicative of the future development of first-in-class inhibitors, such as vaborbactam. Hopefully, insight will be taken from the mechanisms of current inhibitors, both approved and in development, to continue designing and optimizing combinational therapies.

## Figures and Tables

**Figure 1 antibiotics-13-00472-f001:**
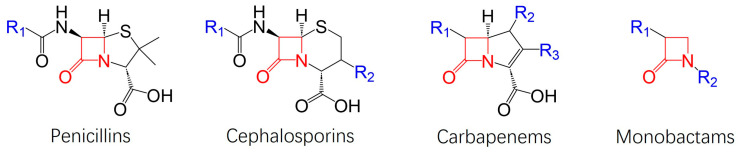
Structural classification of beta-lactams.

**Figure 2 antibiotics-13-00472-f002:**
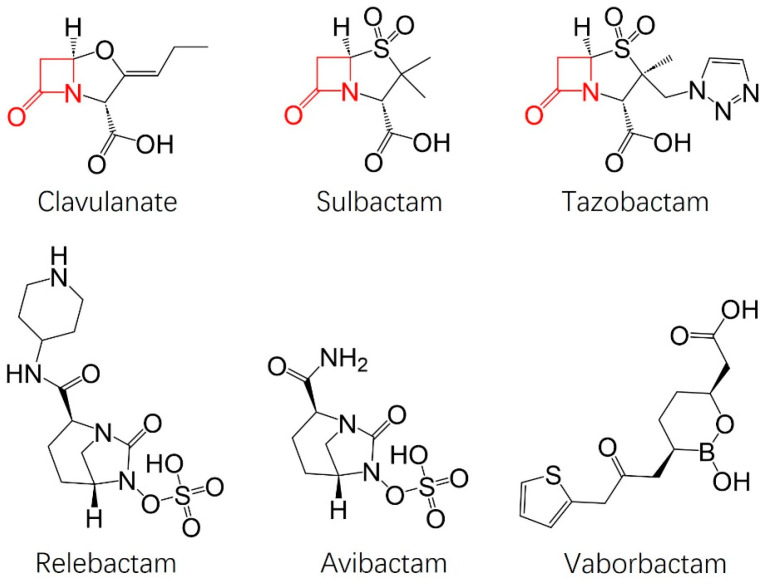
Structure of FDA-approved beta-lactamase inhibitors.

**Figure 3 antibiotics-13-00472-f003:**
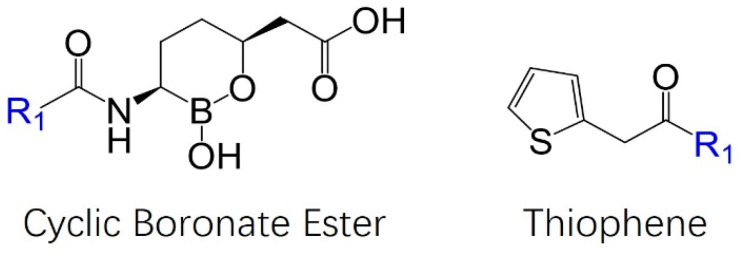
The cyclic boronate ester substructure of Vaborbactam and the thiophene substituent slated for attachment to the R1 group. The addition of the thiophene R group to cyclic boronate ester forms what is now known as vaborbactam.

**Figure 4 antibiotics-13-00472-f004:**
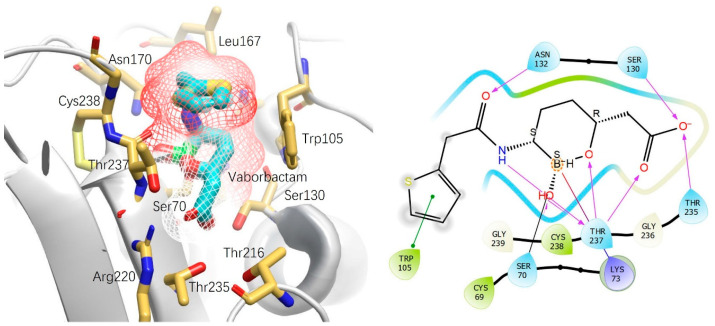
X-ray structure of vaborbactam in complex with KPC-2. Ser70 of KPC-2 (gray) covalently bound to boron in vaborbactam (cyan). Other significant pharmacophores include hydrogen bonding of the acetyl group with Ser130 and Arg220. The amide also contributes to the high affinity of vaborbactam in the oxyanion hole with hydrogen bonding between Thr237. The thiophene utilizes the hydrophobic effect, as seen by the solvent exposure, shielding, and securing vaborbactam to the binding site while also undergoing *pi-pi* stacking with Trp105.

**Figure 5 antibiotics-13-00472-f005:**
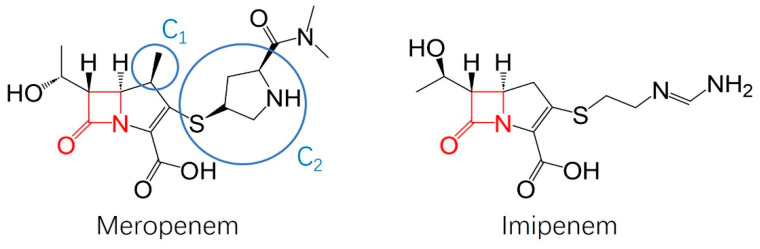
Chemical structure of meropenem and imipenem with structure–activity relationships. From left to right, the key structural differences in meropenem shown in blue circles include the C_1_ methyl group and the C_2_ pyrrolidine-3-thiol group.

**Table 1 antibiotics-13-00472-t001:** Inhibition activity of vaborbactam.

Enzyme	Class	*K_i_* (µM)
KPC-2	A	0.069
KPC-3	A	0.050
CTX-M-14	A	0.033
CTX-M-15	A	0.030
SHV-12	A	0.029
TEM-10	A	0.110
TEM-43	A	1.04
AmpC	C	0.035
P99	C	0.053
CMY-2	C	0.099
OXA-48	D	14
OXA-23	D	66
NDM-1	B	>160
VIM-1	B	>160
References [16,26]		

**Table 2 antibiotics-13-00472-t002:** Physical and chemical properties of meropenem and vaborbactam.

Property	Meropenem	Vaborbactam
LogP	−0.6	1.86
pKa	3.47/9.39	3.75/−2.6
Molecular Weight	383.5 g/mol	297.14 g/mol
Formal Charge	0	0
Solubility in Water	5.63 mg/mL	0.155 mg/mL
H-Bond Donors	3	3
H-Bond Acceptors	7	6
Rotatable Bonds	5	5
References [31,32,33,34]		

**Table 3 antibiotics-13-00472-t003:** Pharmacokinetic data from non-infected patients in phase 1 studies.

Parameter	Meropenem	Vaborbactam
CL_R,max_ (L/h)	6.58	8.86
CL_NR_ (L/h)	3.85	0.157
CL_d_ (L/h)	1.36	2.75
V_c_ (L)	17.0	17.1
V_p_ (L)	2.32	1.77
eGFR_50_ (mL/min/1.73 m^2^)	40.0	49.7
Reference [43]		

CL_R_—renal clearance, CL_NR_—non-renal clearance, CL_d_—distributional clearance, V_c_—central compartment volume, V_p_—peripheral compartment volume, eGFR_50_—eGFR value at half-maximal CL_R_.

**Table 4 antibiotics-13-00472-t004:** Pharmacokinetic data from phase 3 clinical trials.

Parameter	Meropenem	Vaborbactam
C_max_ (µg/mL)	55.4	68.7
Day 1 AUC_0–24_ (µg h/mL)	593	776
Steady-state AUC_0–24_ (µg h/mL)	586	766
CL (L/h)	8.68	6.22
t_1/2,alpha_	0.771	0.379
t_1/2,beta_	1.89	2.04
Reference [43]		

Data pooled from studies 505 and 506. C_max_—highest concentration observed during the first interval.

**Table 5 antibiotics-13-00472-t005:** Adverse events recorded from clinical trials.

Adverse Event	Meropenem	Vaborbactam	Meropenem + Vaborbactam
Headache	0.4	29.2	8.8
Diarrhea	2.5	ND	3.3
Nausea/Lathargy	1.2	20.8	1.8
Rash	1.4	12.5	ND
Injection site reaction	0.9	41.7	2.2
Sepsis	0.1	ND	ND
Reference	[46]	[44]	[45]

Represented as the percentage of the population that experienced the event; ND represents no data.

**Table 6 antibiotics-13-00472-t006:** IC_50_ values of vaborbactam against mammalian serine proteases.

Enzyme	IC_50_ (µM)
Trypsin	>1000
Chymotrypsin	>1000
Plasmin	>1000
Thrombin	>1000
Elastase	>1000
Urokinase	>1000
Tissue plasminogen activator (TPA)	>1000
Chymase	>1000
D-dipeptidyl peptidase 7 (DPP7)	>1000
Neutrophil elastase	>1000
Cathepsin A	1000
Reference [26]	

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
