# Peer review of "Meropenem/Vaborbactam—A Mechanistic Review for Insight into Future Development of Combinational Therapies"

_antibiotics, 2024, doi:10.3390/antibiotics13060472_

Round 1

Reviewer 1 Report

Comments and Suggestions for Authors

The manuscript adeptly contextualizes the review by recognizing the historical significance of beta-lactam antibiotics, underscoring the challenge posed by antibiotic resistance, and introducing combinational therapy as a potential solution. The authors focus specifically on Vabomere, a recently FDA-approved beta-lactam combinational therapy, which adds relevance and immediacy to the discussion, addressing a current development in antibiotic therapy. While the manuscript is well-crafted and effectively presented, there are areas where improvements could be made to maintain the quality of the research article.

1.     There is a need for additional recent citations (post-2021) to support claims made in both the introduction and discussion sections.

2.     The term "In Silico" should be italicized throughout the manuscript, as exemplified in line 85, to adhere to formatting conventions.

3.     The manuscript should underscore the importance of synergistic effects in combination antibiotic therapy within the context of combating antibiotic resistance. This aspect warrants further emphasis to elucidate its significance in overcoming the challenges posed by resistant bacterial strains.

Author Response

We have revised the manuscript considering the Reviewers’ comments. We feel that we were able to satisfactorily address the comments of four Reviewers and have worked to improve the clarity of our writing as suggested by the Reviewers.

Below we detail the revisions made to the manuscript and our responses to the comments of all Reviewers. While hoping for positive feedback, we will be grateful for further comments and suggestions.

Sincerely,

Woo Shik Shin, on behalf of all authors.

Reviewer 1

The manuscript adeptly contextualizes the review by recognizing the historical significance of beta-lactam antibiotics, underscoring the challenge posed by antibiotic resistance, and introducing combinational therapy as a potential solution. The authors focus specifically on Vabomere, a recently FDA-approved beta-lactam combinational therapy, which adds relevance and immediacy to the discussion, addressing a current development in antibiotic therapy. While the manuscript is well-crafted and effectively presented, there are areas where improvements could be made to maintain the quality of the research article.

  1. There is a need for additional recent citations (post-2021) to support claims made in both the introduction and discussion sections.

Answer- Recent citations (after 2021) have been added to support the arguments presented in both the Introduction and Discussion sections.

(56) Tiseo, G., Galfo, V., Riccardi, N. et al. Real-world experience with meropenem/vaborbactam for the treatment of infections caused by ESBL-producing Enterobacterales and carbapenem-resistant Klebsiella pneumoniae. Eur J Clin Microbiol Infect Dis 2024.

(57) Tiseo, G., Suardi, L.R., Leonildi, A., Giordano, C., Barnini, S., Falcone, M. Meropenem/vaborbactam plus aztreonam for the treatment of New Delhi metallo-β-lactamase-producing Klebsiella pneumoniae infections. Journal of Antimicrobial Chemotherapy 2023, 78, 2377–2379.

(58) Cienfuegos-Gallet AV, Shashkina E, Chu T, Zhu Z, Wang B, Kreiswirth BN, Chen L. 2024. In vitro activity of meropenem-vaborbactam plus aztreonam against metallo-β-lactamase-producing Klebsiella pneumoniae. Antimicrob Agents Chemother68:e01346-23.https://doi.org/10.1128/aac.01346-23

  1. The term "In Silico" should be italicized throughout the manuscript, as exemplified in line 85, to adhere to formatting conventions.

Answer- To comply with the manuscript formatting guidelines, "in silico" has been italicized throughout the entire manuscript.

  1. The manuscript should underscore the importance of synergistic effects in combination antibiotic therapy within the context of combating antibiotic resistance. This aspect warrants further emphasis to elucidate its significance in overcoming the challenges posed by resistant bacterial strains.

Answer- The following sentence emphasizing the importance of the synergistic effects of combination antibiotic therapy has been included in the manuscript.

“The development of a new class of potent β-lactamase inhibitors to address the existing β-lactam antibiotic resistance offers the greatest opportunity for maximizing the efficacy of combination antimicrobial therapy, aiming to preserve the potency of existing β-lactam antibiotics.”

Reviewer 2 Report

Comments and Suggestions for Authors

What does "a minute infection" mean in the introduction part (line 21)? Please clarify.

Consider introducing Vabomere earlier in the manuscript, such as in the introduction section, to provide context for its later discussion.

The title of Figure 2 should be about "beta-lactamase inhibitors" not "beta-lactamase". 

Table 5 appears to have two identical columns for Meropenem. Should one of them be Vaborbactam?

Comments on the Quality of English Language

Please fix the typo "Vabormere" (line 149).
Line 225-227: The sentences "Vabomere is a unique combinational therapy being that the structure of its inhibitor is a boronic acid. For this reason, it can be expected to ascendant will new ideas in the structural development of future combinational therapies." contain grammatical errors. Please correct. 

Author Response

We have revised the manuscript considering the Reviewers’ comments. We feel that we were able to satisfactorily address the comments of four Reviewers and have worked to improve the clarity of our writing as suggested by the Reviewers.

Below we detail the revisions made to the manuscript and our responses to the comments of all Reviewers. While hoping for positive feedback, we will be grateful for further comments and suggestions.

Sincerely,

Woo Shik Shin, on behalf of all authors.

Reviewer 2

  1. What does "a minute infection" mean in the introduction part (line 21)? Please clarify.

Answer- Minute was changed to “minor infection”

  1. Consider introducing Vabomere earlier in the manuscript, such as in the introduction section, to provide context for its later discussion.

Answer- the following sentence was added in the Introduction lines 25-27.

“To no surprise, beta-lactams have become the largest class of antibiotics today, covering 65% of the market, and being the inspiration for modern treatments like meropenem/vaborbactam [3].”

  1. The title of Figure 2 should be about "beta-lactamase inhibitors" not "beta-lactamase".

Answer- Thank you for pointing out areas that require correction. The correction has been made in the manuscript.

  1. Table 5 appears to have two identical columns for Meropenem. Should one of them be Vaborbactam?

Please fix the typo "Vabormere" (line 149).

Line 225-227: The sentences "Vabomere is a unique combinational therapy being that the structure of its inhibitor is a boronic acid. For this reason, it can be expected to ascendant will new ideas in the structural development of future combinational therapies." contain grammatical errors. Please correct.

Answer- Thank you for pointing out areas that require correction. Typo was fixed in line 149 and lines 225-227.

Reviewer 3 Report

Comments and Suggestions for Authors

In the manuscript entitled "Vabomere - A Mechanistic Review for Insight on Future Development of Combinational Therapies", the development of Vabomere (Vaborem), a meropenem/vaborbactam beta-lactam combinational therapy, was reviewed in terms of structure rationale, activity gamut, pharmacodynamic/pharmacokinetic properties, and toxicity to provide insight into the future development of analogous therapies. However, the content of the manuscript is not up-to-date. The recently published works are lacking. The brand name of "Vabomere" has been changed as "Vaborem", even this information is lacking in the manuscript. Instead of the brand name, utilization of "meropenem/vaborbactam" is preferred. The abstract could be ended with a conclusion sentence.

Author Response

We have revised the manuscript considering the Reviewers’ comments. We feel that we were able to satisfactorily address the comments of four Reviewers and have worked to improve the clarity of our writing as suggested by the Reviewers.

Below we detail the revisions made to the manuscript and our responses to the comments of all Reviewers. While hoping for positive feedback, we will be grateful for further comments and suggestions.

Sincerely,

Woo Shik Shin, on behalf of all authors.

Reviewer 3

  1. In the manuscript entitled "Vabomere - A Mechanistic Review for Insight on Future Development of Combinational Therapies", the development of Vabomere (Vaborem), a meropenem/vaborbactam beta-lactam combinational therapy, was reviewed in terms of structure rationale, activity gamut, pharmacodynamic/pharmacokinetic properties, and toxicity to provide insight into the future development of analogous therapies. However, the content of the manuscript is not up-to-date. The recently published works are lacking. The brand name of "Vabomere" has been changed as "Vaborem", even this information is lacking in the manuscript. Instead of the brand name, utilization of "meropenem/vaborbactam" is preferred. The abstract could be ended with a conclusion sentence.

Answer- Thank you for bringing up this important point. "Vabomere" has been replaced with "meropenem/vaborbactam" throughout the manuscript.

Reviewer 4 Report

Comments and Suggestions for Authors

The development of Vabomere - a recently FDA-approved beta-lactam combinational therapy - is reviewed in terms of structure rationale, activity gamut, pharmacodynamic/pharmacokinetic properties, and toxicity to provide insight into the future development of analogous therapies. The review is clear, and relevant, in context of the increase in bacterial antibiotic resistance. In general, the review provides elements to integrate structural aspects that facilitate interactions between the active sites of b Lactamases and the inhibitors used in combinational therapies and to compare the effectiveness of different variants in each class of B-lactam antibiotics.

There are recent reviews, some of them include studies on the Safety, Tolerability, and Pharmacokinetics of Vaborbactam and Meropenem Alone and in Combination (cited in the manuscript on reference 42), or describe  new  β-Lactam-β-Lactamase inhibitor combinations (Fong, I.W. (2023). New β-Lactam-β-Lactamase Inhibitor Combinations. In: New Antimicrobials: For the Present and the Future. Emerging Infectious Diseases of the 21st Century. Springer, Cham. https://doi.org/10.1007/978-3-031-26078-0_3) or report the efficacy of combination therapies  (Umemura, T.; Kato, H.; Hagihara, M.; Hirai, J.; Yamagishi, Y.; Mikamo, H. Efficacy of Combination Therapies for the Treatment of Multi-Drug Resistant Gram-Negative Bacterial Infections Based on Meta-Analyses. Antibiotics 202211, 524. https://doi.org/10.3390/antibiotics11040524), or present information to discover new combinations to overcome antibiotic resistance (Zhu M, Tse MW, Weller J, Chen J, Blainey PC. The future of antibiotics begins with discovering new combinations. Ann N Y Acad Sci. 2021 Jul;1496(1):82-96. doi: 10.1111/nyas.14649. Epub 2021 Jul 2. PMID: 34212403; PMCID: PMC8290516).

The main contribution of the present review is the integration of the elements considered in the reviews that have been published, adding the aspects related to the structural interactions with the target entities.

The cited references are relevant, but mostly are not recent publications (within the last 5 years), 22 references were published before 2014 (reference 4, 6, 8, 10, 11, 14, 19, 24, 25, 27, 29, 30, 35, 37, 39, 46, 47, 48, 49, 50, 53 and 54). It is suggested to include recent publications to have certainty of timeliness of the contents.

Reference 12 is incomplete, te complete one is:  Benin, B.M.; Hillyer, T.; Shin, W.S.. Quercetin, A Potential Metallo-β-Lactamase Inhibitor for Use in Combination Therapy Against β-Lactam Antibiotic-Resistant Bacteria. Open J Pathol Toxicol Res. 1(1): 2021. OJPTR.MS.ID.000503.

In references 17, 18, 42, 43 AND 44  the year of publication is missing.

It is not clear what does reference 40 express.

Figures most be improved, Figure 1 present a mistake in the structure of Cephalosporins (N symbol is missing in lactam ring.

Figure 3 is confusing, it would be preferable to say that R1 is the thiophene illustrated, rather than to assume that R1 is bound to thiophene.

Author Response

Reviewer 4

The development of Vabomere - a recently FDA-approved beta-lactam combinational therapy - is reviewed in terms of structure rationale, activity gamut, pharmacodynamic/pharmacokinetic properties, and toxicity to provide insight into the future development of analogous therapies. The review is clear, and relevant, in context of the increase in bacterial antibiotic resistance. In general, the review provides elements to integrate structural aspects that facilitate interactions between the active sites of b Lactamases and the inhibitors used in combinational therapies and to compare the effectiveness of different variants in each class of B-lactam antibiotics.

There are recent reviews, some of them include studies on the Safety, Tolerability, and Pharmacokinetics of Vaborbactam and Meropenem Alone and in Combination (cited in the manuscript on reference 42), or describe  new  β-Lactam-β-Lactamase inhibitor combinations (Fong, I.W. (2023). New β-Lactam-β-Lactamase Inhibitor Combinations. In: New Antimicrobials: For the Present and the Future. Emerging Infectious Diseases of the 21st Century. Springer, Cham. https://doi.org/10.1007/978-3-031-26078-0_3) or report the efficacy of combination therapies  (Umemura, T.; Kato, H.; Hagihara, M.; Hirai, J.; Yamagishi, Y.; Mikamo, H. Efficacy of Combination Therapies for the Treatment of Multi-Drug Resistant Gram-Negative Bacterial Infections Based on Meta-Analyses. Antibiotics 2022, 11, 524. https://doi.org/10.3390/antibiotics11040524), or present information to discover new combinations to overcome antibiotic resistance (Zhu M, Tse MW, Weller J, Chen J, Blainey PC. The future of antibiotics begins with discovering new combinations. Ann N Y Acad Sci. 2021 Jul;1496(1):82-96. doi: 10.1111/nyas.14649. Epub 2021 Jul 2. PMID: 34212403; PMCID: PMC8290516).

The main contribution of the present review is the integration of the elements considered in the reviews that have been published, adding the aspects related to the structural interactions with the target entities.

  1. The cited references are relevant, but mostly are not recent publications (within the last 5 years), 22 references were published before 2014 (reference 4, 6, 8, 10, 11, 14, 19, 24, 25, 27, 29, 30, 35, 37, 39, 46, 47, 48, 49, 50, 53 and 54). It is suggested to include recent publications to have certainty of timeliness of the contents.

Answer- To ensure the timeliness of the content, the following recent reference has been added to the manuscript.

(56) Tiseo, G., Galfo, V., Riccardi, N. et al. Real-world experience with meropenem/vaborbactam for the treatment of infections caused by ESBL-producing Enterobacterales and carbapenem-resistant Klebsiella pneumoniae. Eur J Clin Microbiol Infect Dis 2024.

(57) Tiseo, G., Suardi, L.R., Leonildi, A., Giordano, C., Barnini, S., Falcone, M. Meropenem/vaborbactam plus aztreonam for the treatment of New Delhi metallo-β-lactamase-producing Klebsiella pneumoniae infections. Journal of Antimicrobial Chemotherapy 2023, 78, 2377–2379.

(58) Cienfuegos-Gallet AV, Shashkina E, Chu T, Zhu Z, Wang B, Kreiswirth BN, Chen L. 2024. In vitro activity of meropenem-vaborbactam plus aztreonam against metallo-β-lactamase-producing Klebsiella pneumoniae. Antimicrob Agents Chemother68:e01346-23.https://doi.org/10.1128/aac.01346-23

  1. Reference 12 is incomplete, te complete one is: Benin, B.M.; Hillyer, T.; Shin, W.S.. Quercetin, A Potential Metallo-β-Lactamase Inhibitor for Use in Combination Therapy Against β-Lactam Antibiotic-Resistant Bacteria. Open J Pathol Toxicol Res. 1(1): 2021. OJPTR.MS.ID.000503.

Answer- The citation was updated

  1. In references 17, 18, 42, 43 AND 44 the year of publication is missing.

Answer- Publication year was added to the mentioned references

  1. It is not clear what does reference 40 express.

Answer- Reference 40 is a drug data sheet from the FDA. The URL was added to the reference for access

  1. Figures most be improved, Figure 1 present a mistake in the structure of Cephalosporins (N symbol is missing in lactam ring.

Answer- The structural defect of the compound has been corrected.

  1. Figure 3 is confusing, it would be preferable to say that R1 is the thiophene illustrated, rather than to assume that R1 is bound to thiophene.

Answer- To avoid confusion, the following sentence has been replaced with the one below for clarity.

“The cyclic boronate ester substructure of Vaborbactam and the thiophene substituent slated for attachment to the R1 group.”

Round 2

Reviewer 3 Report

Comments and Suggestions for Authors

The issues raised in the earlier version of the manuscript have mostly been addressed in the revised version. The number of recent articles could be increased as references. The final version of the manuscript can be accepted for publication.